# Scattering and Directionality Effects of Noise Generation from Flapping Thrusters Used for Propulsion of Small Ocean Vehicles

Kostas Belibassakis [1], John Prospathopoulos [2,*] and Iro Malefaki [1]

1   School of Naval Architecture & Marine Engineering, National Technical University of Athens (NTUA), Heroon Polytechniou 9, 15780 Athens, Greece
2   School of Mechanical Engineering, National Technical University of Athens (NTUA), Heroon Polytechniou 9, 15780 Athens, Greece
*   Correspondence: jprosp@fluid.mech.ntua.gr; Tel.: +30-2107721048

**Abstract:** Flapping-foil thrusters are systems that operate at a substantially lower frequency compared with marine propellers and are characterized by a much smaller power concentration. These biomimetic devices are able to operate very efficiently, offering desirable levels of thrust required for the propulsion of small vessels or autonomous underwater vehicles (AUVs), and can be used for the standalone propulsion of small vessels or for augmenting ship propulsion in waves, alleviating the generation of noise and its adverse effects on sea life, particularly on marine mammals. In this work, we consider the generation of noise by flapping foils arranged in the neighborhood of the above vessels including the scattering effects by the hull, which, in addition to free-surface and seabed effects, significantly contribute to the modification of the characteristics of the acoustic field. A Boundary Element Method (BEM) is developed to treat the 3D scattering problem in the frequency domain forced by monopole and dipole source terms associated with the Ffowcs Williams and Hawkings (FW-H) equation. Numerical results are presented in selected cases illustrating that the hull geometry and acoustic properties, as well as the sea surface and seabed effects, are important for the determination of the directionality of the generated noise and significantly affect the propagation in the underwater ocean environment.

**Keywords:** flapping thruster; AUV; noise generation and propagation; scattering and directionality effects

## 1. Introduction

In the last decades, the seas have become substantially noisier and anthropogenic sources contribute considerably to this degradation trend, with detrimental effects on sea life, particularly on marine mammals (e.g., [1]). Shipping operations' resource and infrastructure development have increased the noise generated by human activities, whereas sounds of biological origin have been reduced by hunting, fishing, and habitat degradation. In particular, ambient ocean noise levels below 300 Hz have increased by 15–20 dB over the last century, and shipping noise contributes significantly to such increases in this frequency range [2].

Many recent studies have shown that underwater-radiated noise from commercial ships may have both short- and long-term negative consequences on sea life. The issue of underwater noise and its impact on marine mammals was first raised by the IMO in 2004. It was noted that continuous anthropogenic noise in the ocean was primarily generated by shipping. Since ships routinely cross international boundaries, the management of such noise required a coordinated international response. Moreover, in 2008, the IMO Marine Environment Protection Committee (MEPC) [3,4] agreed to develop non-mandatory technical guidelines to minimize the introduction of incidental noise from commercial

shipping operations into the marine environment to reduce the potential adverse impacts on marine life.

As far as the radiated noise is concerned, it has been found that different components are dominant at different speeds. In particular, hydrodynamic noise due to propeller operation in the wake of the ship and machinery is dominant at low speeds, whereas propeller noise is dominant at higher speeds, especially when cavitation takes place [5]. Marine propellers are the standard devices used for ship propulsion and operate at high rotational speeds so that their blades produce the required forward thrust. High-flow velocity increases the likelihood of cavitation, and partial cavitation usually appears near the tip region and occasionally also at the hub of marine propeller blades and the trailing vortex sheets. The fast variation of the generated bubble cavitation volume on the propeller blades, acting as acoustic monopole terms, in conjunction with the dipole contribution due to unsteady blade loading, leads to the generation of intensive noise, especially at the blade frequency and the first harmonics, while at higher frequencies, noise is caused by sheet cavity collapse and shock wave generation [6]. The lower frequency band of the noise generated by marine propellers, especially when they operate under partial cavitating conditions, has a negative impact on the life conditions of marine mammals. Additional discussion concerning underwater noise from marine propellers including the latest research advances can be found in several sources—see, for e.g., [7,8] and the references therein.

On the other hand, flapping-foil thrusters are systems operating at a substantially lower frequency compared with marine propellers and are characterized by much smaller power concentrations. These biomimetic devices are able to operate very efficiently while offering the desirable levels of thrust required for the propulsion of small vessels or autonomous underwater vehicles (AUVs); for examples, see [9,10]. An extended review of hydrodynamic scaling laws in aquatic locomotion and fishlike swimming can be found in [11]. Moreover, flapping-foil configurations have been investigated both as main propulsion devices and for augmenting ship propulsion in waves, substantially improving the performance by the exploitation of renewable wave energy. More details can be found in [12,13], as well as a review in [14]. In the framework of the Seatech H2020 project entitled "Next generation short-sea ship dual-fuel engine and propulsion retrofit technologies" (https://seatech2020.eu/, accessed on 5 August 2022), a concept of symbiotic ship engine and propulsion innovations is studied that, when combined, are expected to lead to a significant increase in fuel efficiency and a reduction in greenhouse gas emissions. The proposed renewable energy-based propulsion innovation is based on the biomimetic dynamic wing, mounted at the ship bow to augment the ship's propulsion in moderate and higher sea states, capturing wave energy and producing extra thrust while damping ship motions.

In this work, we consider the generation of noise by flapping foils arranged in the neighborhood of the above vessels, including the scattering effects by the hull, which, in addition to free-surface and seabed effects, significantly contribute to the modification of the characteristics of the acoustic field. A boundary element method (BEM) is developed to treat the three-dimensional (3D) scattering problem in the frequency domain forced by monopole and dipole source terms associated with the Ffowcs Williams and Hawkings (FW-H) [15] equation. Numerical results are presented in selected cases illustrating that the hull geometry and acoustic properties, as well as the sea surface and seabed effects, are important for the determination of the directionality of the generated noise and significantly affect the propagation in the underwater ocean environment.

## 2. Noise Generation from Flapping Thruster

The flapping thruster operates as an unsteady hydrofoil in combined oscillatory heaving motion $h(t) = h_0 \sin(\omega t)$ and pitching motion $\theta(t) = \theta_0 \sin(\omega t + 0.5\pi)$ with a phase difference of about 90 degrees, as depicted in Figure 1. The most important non-dimensional parameters are the Strouhal number $Str = 2fh_0/U$, where $f = \omega/2\pi$ is the frequency, the heaving amplitude $h_0$, the forward travelling speed $U$, the heave-to-chord

ratio $h_0/c$, the pitching amplitude $\theta_0$, and the phase difference between the foil heave and pitch oscillatory motions.

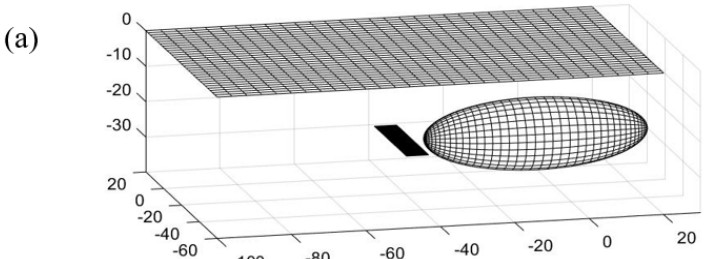

(a)

(b)

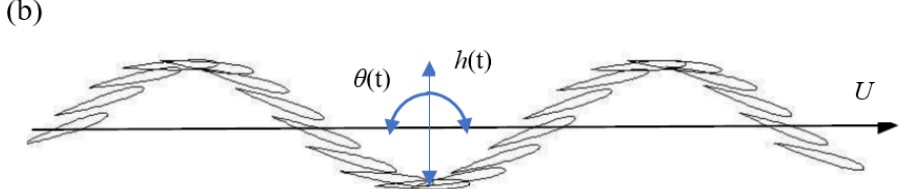

**Figure 1.** (**a**) Flapping thruster used for the propulsion of an AUV or small ocean vehicle. (**b**) Consecutive positions of foil oscillatory motion due to combined heaving $h(t)$ and pitching $\theta(t)$ motion.

The unsteady forces lead to the generation of oscillatory vertical lift force $F_z(t)$ and horizontal thrust force $F_x(t)$ with a significant non-zero mean (blue arrow in Figure 1). The time history of the total foil force is given from Equation (1):

$$\mathbf{F}(t) = F_x(t)\mathbf{i} + F_z(t)\mathbf{k} \tag{1}$$

where $\mathbf{i}, \mathbf{k}$ are the horizontal and vertical unit vectors, respectively. The calculation of hydrodynamic loads can be obtained by the pressure integration on the foil using various methods, as, for example, those used by [16–18]. It is noted that, even in the single frequency case of flapping-foil kinematics, the hydrodynamic forces are periodic, but contain multiple harmonics due to the effects of various nonlinearities and the fact that the horizontal thrust force component $F_x(t)$ is manifested at the double frequency.

Low-frequency noise caused by the fluctuations of foil pressure and volume flow disturbance due to the oscillatory motion of the flapping thruster can be predicted in free space using the Ffowcs Williams and Hawkings equation (FW-H), described as follows:

$$\frac{1}{c^2}\partial_t^2 p(\mathbf{x}, t) - \nabla^2 p(\mathbf{x}, t) = f_m + f_d + f_q \tag{2}$$

where $c$ is the speed of sound in the medium ($c$ ranges from 1500 to 1550 m/s for seawater), while the various terms on the right-hand side correspond to the acoustic monopole $f_m$, dipole $f_d$, and quadrupole source terms $f_q$, respectively [19]. The quadrupole term is predominantly associated with the turbulence and vorticity-induced noise or strongly transonic flow phenomena and becomes important at higher frequencies. Focusing on the low-frequency part of the generated noise spectrum, the contributions from the latter are neglected in the present work. Following Farassat [20], the formulation of an integral representation of the solution of Equation (2) forced by the monopole and dipole terms is considered. Taking into account that the speed of sound in water is much greater than the flow velocities, the following approximation, concerning the noise generation by the flapping thruster from the monopole and dipole sources, is obtained (see also [16]):

$$p_m(\mathbf{x}, t) \approx \frac{\rho}{4\pi} \frac{d^2 Q(t_r)}{dt^2} \frac{1}{|\mathbf{x} - \mathbf{x}_Q(t_r)|} \tag{3}$$

$$p_d(\mathbf{x}, t) \approx -\frac{1}{4\pi c} \frac{d\mathbf{F}(t_r)}{dt} \frac{\mathbf{x} - \mathbf{x}_T(t_r)}{r^2} + \frac{1}{4\pi} \mathbf{F}(t_r) \frac{\mathbf{x} - \mathbf{x}_T(t_r)}{r^3} \tag{4}$$

where $p_m$, $p_d$ is the acoustic pressure from the monopole and dipole sources respectively, $t_r = r/c$ denotes the retarded time between the observation point $\mathbf{x}$ in the foil frame of reference and the hydrodynamic force considered to be applied at the hydrodynamic pressure center of the foil $\mathbf{x}_T$, and $\mathbf{x}_Q$ denotes the center of volume displaced by the foil. In the case of an unsteady cavitating foil thruster, the latter term corresponds mainly to the bubble cavitation volume. Figure 2 presents two different time snapshots of the acoustic pressure field obtained from Equation (4) in the case of a flapping thruster operating in water. The hydrodynamic foil loads have been calculated by pressure integration using the results obtained by the 3D BEM method of [16], which is briefly presented in the Appendix A.

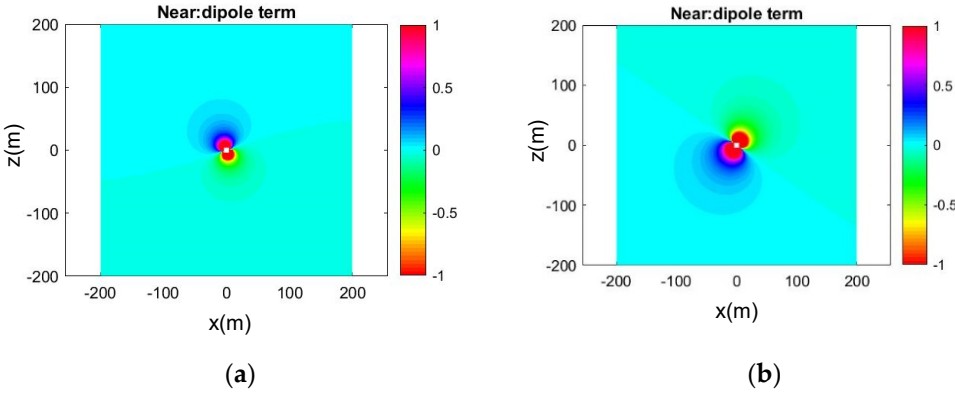

(**a**)                                                       (**b**)

**Figure 2.** Time snapshots of acoustic field generated by dipole sources in the case of flapping thruster operating in water ($c$ = 1500 m/s), in the case of foil with NACA0012 sections of Figure 1 flapping at $Str$ = 0.23, $h_0/c$ = 0.75, $\theta_0$ = 23 deg, using the calculated hydrodynamic loads from a pressure integration 3D BEM method. (**a**) real part, (**b**) imaginary part.

The periodic time-series of acoustic force $\mathbf{F}(t)$ and cavity volume data $Q(t)$ can be represented by Fourier series in the form:

$$Q(t) = Q_0^c + \sum_{n=1} Q_n^c \cos(n\omega t) + Q_n^s \sin(n\omega t) \tag{5}$$

$$F_x(t) = X_0^c + \sum_{n=1} X_n^c \cos(n\omega t) + X_n^s \sin(n\omega t) \tag{6}$$

$$F_z(t) = Z_0^c + \sum_{n=1} Z_n^c \cos(n\omega t) + Z_n^s \sin(n\omega t) \tag{7}$$

including the basic harmonic ($n$ = 1) as well as its multiples. By converting Equation (2) to the frequency domain, assuming $p(\mathbf{x}, t) = \text{Re}\left(\sum_{n=1} p(\mathbf{x}; \omega_n) e^{-i\omega_n t}\right)$ and focusing on a single harmonic (basic frequency or its multiples), the above equation takes the form:

$$\nabla^2 p(\mathbf{x}; \omega_n) + k_n^2 p(\mathbf{x}; \omega_n) = Q_n \, \delta(\mathbf{x} - \mathbf{x}_0) + X_n \frac{\partial}{\partial x}\delta(\mathbf{x} - \mathbf{x}_0) + Z_n \frac{\partial}{\partial z}\delta(\mathbf{x} - \mathbf{x}_0) \tag{8}$$

where $k_n = n\omega/c$, and $\mathbf{x}_0$ now stands for the center of volume or the center of pressure. Using the expression of the Green's function of the Helmholtz equation in free space,

$$G(\mathbf{x}, \mathbf{x}_0) = \frac{1}{4\pi} \frac{e^{ik_n|\mathbf{x} - \mathbf{x}_0|}}{|\mathbf{x} - \mathbf{x}_0|} \tag{9}$$

and its derivative

$$\nabla_{x_0} G(\mathbf{x}, \mathbf{x}_0) = \frac{e^{ik_n|\mathbf{x}-\mathbf{x}_0|}}{4\pi|\mathbf{x}-\mathbf{x}_0|^2} \left( \frac{(\mathbf{x}-\mathbf{x}_0)}{|\mathbf{x}-\mathbf{x}_0|} - ik_n(\mathbf{x}-\mathbf{x}_0) \right) \tag{10}$$

the expressions provided by Equations (3) and (4) are the time-domain equivalent of the solution of the Helmholtz equation with complex monopole and dipole source intensities provided by:

$$Q_n = -\rho n^2 \omega^2 (Q_n^c - iQ_n^s), \ X_n = X_n^c - iX_n^s, \ Z_n = Z_n^c - iZ_n^s \tag{11}$$

that is:

$$p_m(\mathbf{x},t) = \frac{1}{4\pi}\text{Re}\left( \sum_n Q_n \frac{e^{ik_n|\mathbf{x}-\mathbf{x}_0|}}{|\mathbf{x}-\mathbf{x}_0|} e^{-i\omega_n t} \right) \tag{12}$$

$$p_d(\mathbf{x},t) = \frac{1}{4\pi}\text{Re}\left( \sum_n (X_n\mathbf{i} + Z_n\mathbf{k}) \frac{e^{ik_n|\mathbf{x}-\mathbf{x}_0|}}{4\pi|\mathbf{x}-\mathbf{x}_0|^2} \left( \frac{(\mathbf{x}-\mathbf{x}_0)}{|\mathbf{x}-\mathbf{x}_0|} - ik_n(x-\mathbf{x}_0) \right) \right) \tag{13}$$

Given the intensity of the complex monopole and dipole source terms at the basic frequency or its multiples, as provided by the hydrodynamic flapping-foil responses described above, in the sequel, the scattering problem is considered, excited by any monopole or dipole term in the vicinity of a 3D body representing the AUV or seagoing vessel, also accounting for the effects of the free-surface and seabed boundary of the ocean acoustic waveguide. For this purpose, first, the expressions of the Green's function and its derivatives, corresponding to Equations (9) and (10), in a plane-horizontal waveguide confined between a pressure-release upper boundary (sea surface) and the lower seabed boundary, will be provided. The latter will be subsequently used to formulate the 3D scattering problem in the ocean acoustic waveguide and to study the scattering and directionality effects of noise generation from flapping thrusters in the sea environment.

## 3. Green's Function in the Ocean Acoustic Waveguide

For simplicity, a homogeneous ocean acoustic waveguide is considered, confined by an acoustically soft boundary and an idealized seabed by an acoustically hard boundary. For any given value of the wavenumber parameter $k_n = n\omega/c$ (denoted in the sequel simply by $k$), the Green's function $G(\mathbf{x}, \mathbf{x}_0)$, $\mathbf{x} = (x, y, z)$, $\mathbf{x}_0 = (x_0, y_0, z_0)$, of the Helmholtz equation, satisfying the free-surface condition at $z = 0$ and hard seabed boundary condition at $z = -h$, is defined by:

$$\nabla^2 G(\mathbf{x}, \mathbf{x}_0) + k^2 G(\mathbf{x}, \mathbf{x}_0) = \delta(\mathbf{x} - \mathbf{x}_0) \tag{14}$$

$$G(\mathbf{x}, \mathbf{x}_0) = 0, \ z = 0 \tag{15}$$

$$\frac{\partial G(\mathbf{x}, \mathbf{x}_0)}{\partial z} = 0, \ z = -h \tag{16}$$

where $\delta(\mathbf{x} - \mathbf{x}_S)$ is the Dirac delta function and z stands for the vertical axis pointing upwards.

### 3.1. Normal-Mode Series

The monopole source field is obtained by the separation of variables, as follows [21]:

$$G(\mathbf{x}, \mathbf{x}_0) = -\frac{i}{4}\sum_{m=1}^{\infty} \widetilde{Z}_m(z_s)\widetilde{Z}_m(z) H_0^{(1)}(k_{rm}r) \tag{17}$$

where the normalized vertical eigenfunctions are:

$$\widetilde{Z}_n(z) = Z_n(z)/\|Z_n\|, \ \int_{-h}^{0} \widetilde{Z}_m(z)\widetilde{Z}_n(z)dz = \delta_{nm} \tag{18}$$

where $\delta_{nm}$ is the Kronecker delta, and are obtained as solutions to the corresponding Vertical Eigenvalue Problem:

$$\frac{d^2 Z_m(z)}{dz^2} + \left[\frac{\omega^2}{c^2} - k_{rm}^2\right] Z_m(z) = 0, \ Z_m(z=0) = 0, \ \frac{dZ_m(z=-h)}{dz} = 0 \quad (19)$$

The solution for $c$ = const is easily obtained in a cylindrical coordinate system $\left\{r^2 = (x-x_0)^2 + (y-y_0)^2, \ \theta = \tan^{-1}((y-y_0)/(x-x_0)), \ z\right\}$ in the form of normal-mode series expansion (see also [21])

$$G(r,z) = -\frac{i}{2h}\sum_{m=1}^{\infty} \sin(k_{zm} z_s)\sin(k_{zm} z)H_0^{(1)}(k_{rm}r) \quad (20)$$

where $k_{zm} = (m-0.5)\pi/h$, $k_{rm} = \sqrt{k^2 - k_{zm}^2}$, $m = 1, 2, \ldots$, are the vertical and horizontal wavenumbers, respectively, the normalized vertical eigenfunctions are $\widetilde{Z}_m(z) = \sqrt{2/h}\sin(k_{zm} z)$, and $H_0^{(1)}(k_{rm}r)$ is the Hankel function of the first kind.

The corresponding expressions modeling the dipole source terms in the acoustic waveguide associated with the noise generation from the lift and thrust foil forces are defined as solutions of:

$$\nabla^2 G_{l,m,n} + k^2 G_{l,m,n} = \frac{\partial^{l+m+n}}{\partial x^l \partial y^m \partial z^n}\delta(\mathbf{x} - \mathbf{x}_0) \quad (21)$$

and are obtained as:

$$G_{l,m,n}(\mathbf{x}, \mathbf{x}_0) = (-1)^{l+m+n}\frac{\partial^{l+m+n}}{\partial x_0^l \partial y_0^m \partial z_0^n}G(\mathbf{x}, \mathbf{x}_0) \quad (22)$$

(see [22]). In the case of the sound field generated by the horizontal dipole (associated with the foil thrust forcing) and vertical dipole (associated with the lift force), the corresponding expressions of the acoustic field are provided as follows:

$$G_X = \frac{\partial G(\mathbf{x}, \mathbf{x}_0)}{\partial x_0} = -\frac{i}{4}\sum_{m=1}^{\infty}\widetilde{Z}_m(z_0)\widetilde{Z}_m(z)k_{rm}H_1^{(1)}(k_{rm}r)\frac{\partial r}{\partial x_0} = \frac{i}{4}\sum_{m=1}^{\infty}\widetilde{Z}_m(z_0)\widetilde{Z}_m(z)H_1^{(1)}(k_{rm}r)\cos\theta \quad (23)$$

$$G_Z = \frac{\partial G(\mathbf{x}, \mathbf{x}_0)}{\partial z_0} = -\frac{i}{4}\sum_{m=1}^{\infty}\frac{d\widetilde{Z}_m(z_0)}{dz}\widetilde{Z}_m(z)H_0^{(1)}(k_{rm}r) \quad (24)$$

where $\frac{\partial r}{\partial x_0} = -\frac{(x-x_0)}{r} = \cos\theta$, and $\theta$ is the azimuthal angle.

The normal mode series expansion of the Green's function, Equation (20), and its derivatives Equations (23) and (24), corresponding to the field excited by the monopole and the horizontal and vertical dipole sources, respectively, in the planar acoustic waveguide are appropriate for the calculation at some distance away from the source point $\mathbf{x} = \mathbf{x}_0$, by truncating the infinite series and keeping the propagating modes and a number of evanescent modes that are necessary for the numerical convergence. In the isovelocity environment $c$ = const considered, the propagating modes correspond to the mode index:

$$m \le M_p = [0.5 + (kh/\pi)] \quad (25)$$

where the brackets denote the integer part; see also [21]. On the contrary, consideration of the whole infinite series is required in order to represent adequately the Green's function in the vicinity of the singularity, which is of utmost importance for the formulation and solution by means of the boundary integral equation methods of the scattering problems associated with the presence of finite bodies in the domain as AUVs and marine vehicles that are considered in this work. For this purpose, an alternative representation is derived in the

next subsection based on multiple images, which is suitable for the accurate representation in the vicinity of the singularity and the implementation of the 3D BEM for the acoustic scattering problem in the waveguide.

### 3.2. Multiple Image Series

According to the multiple image method, the free-space Green's function corresponding to a point monopole source or its derivatives, provided by Equations (9) and (10) respectively, is supplemented by mirror terms located at symmetrical positions with respect to the free-surface ($z = 0$) and seabed surface ($z = -h$) boundary; see [23]. The mirror sources with respect to the free-surface are considered with an opposite sign in order to fulfill the homogeneous Dirichlet boundary condition for an acoustically soft boundary, while the mirror sources with respect to the seabed plane are considered with the same sign in order to satisfy a homogeneous Neumann condition on the hard bottom boundary. The repetitive mirroring process generates an infinite series of simple source terms, which, similar to the case of normal-mode series, can be truncated, keeping the first number of terms. This is justified by the fact that as the position of mirror sources becomes greater, the contribution to field points in the acoustic waveguide near the monopole source and in intermediate distances becomes insignificant. As expected, the above truncated multiple image series provides an excellent representation of the Green's function in the region of the source and for intermediate distances in the near field, and becomes more approximate moving in the far field, requiring more and more terms for convergent results. On the other hand, in the intermediate and far-field region, the representation of the acoustic field in the waveguide by the truncated normal-mode series, discussed in the previous subsection, is excellent. There is an intermediate, quite extended, region where the two representations provide perfectly matched results. In this sense, the multiple image series will be used in the near field for solving the 3D acoustic scattering by the body (AUV or vessel) in the waveguide and the normal-mode series will be used to propagate the acoustic field in the region far from the monopole or dipole sources, representing the excitation from the foil thruster and the body generating the 3D scattering effects.

In order to better represent the derivation of the multiple image series for the Green's function in the acoustic waveguide of depth $h$, consider the original source positioned at $z = z_s < 0$, which is mirrored with respect to the free surface ($z = 0$) and the bottom ($z = -h$), as depicted in Figure 3, where the origin is at the free surface and the z-axis is pointing upwards. When mirroring is made with respect to the free surface, the sign of the field contribution changes to satisfy the Dirichlet condition of zero pressure. When mirroring is made with respect to the bottom, the sign of the field contribution remains in the same in order to satisfy the Neumann condition of zero acoustic velocity. Successive mirroring of the image sources is made until their number is sufficient to properly represent the effect of the sea surface and its bottom. As an example, concerning the first terms in the series, the image sources above the free surface are located at points $\{z = -z_s, 2h + z_s, 2h - z_s, \ldots\}$, with intensities $\{-1, -1, +1, \ldots\}$, and the image sources below the hard seabed are located at points $\{z = -2h - z_s, -2h + z_s, \ldots\}$, with intensities $\{+1, -1, \ldots\}$, as shown in Figure 3. The image series method accurately represents the near field and especially the singularity in the vicinity of the source point. In the far field, the normal mode representation is easier to implement because the image method needs an increasing number of image sources in order to achieve an accurate representation.

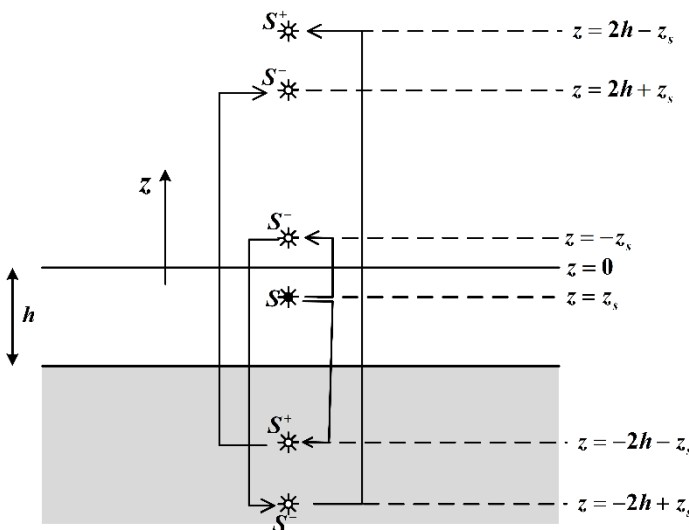

**Figure 3.** Image source method. The original source $S$ is positioned at $z = z_s < 0$.

As an example, Figure 4 depicts the calculated acoustic field excited by the monopole, horizontal dipole, and vertical dipole sources, respectively, of frequency 5 Hz, located at a submergence depth of $z_0 = -300$ m, in an isovelocity $c = 1500$ m/s ocean acoustic waveguide of depth $h = 3000$ m. In this case, the number of propagating modes from Equation (25) is $M_p = 20$ and the acoustic field calculated by the normal mode series is obtained keeping 40 terms in the series Equation (20), and is plotted in the left subplots. For comparison, in the right subplots, the same field obtained from the mirror series using the above first six terms is presented. It is observed in these plots that, except the vicinity of the source (at submergence depth $z_0$ near the origin), the two series expansions of the monopole and dipole source fields provide the same result, up to a horizontal distance of $|x| = 3000$ m, after which the normal mode series provide accurate results. On the contrary, it is the image series that provide accurate results in the whole intermediate region $|x| < 3000$ m, including the excellent representation of the point singularity in the near-source field region $|x| < 300$ m.

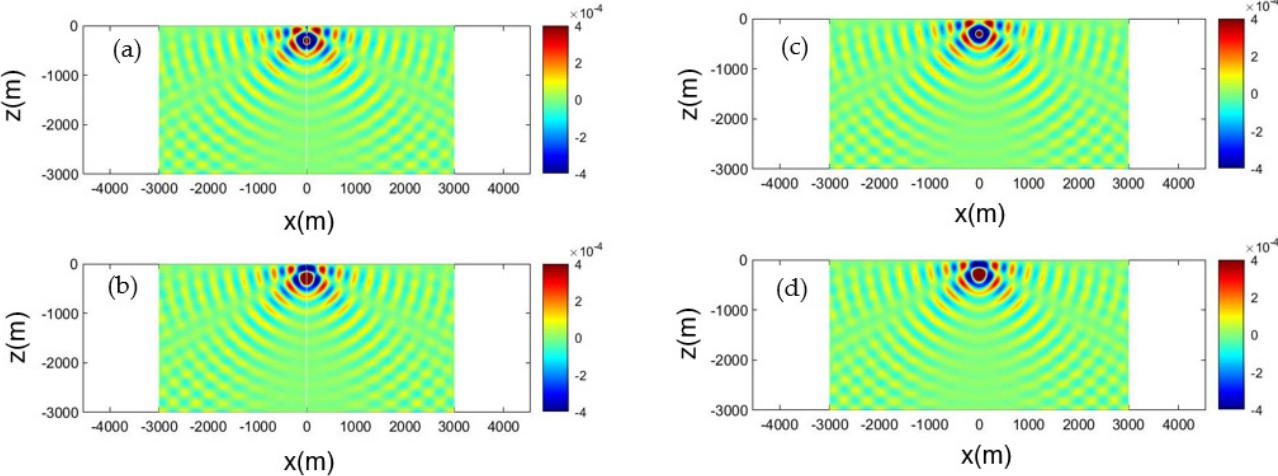

**Figure 4.** Calculated acoustic field corresponding to the monopole source, Equation (20). Source frequency 5 Hz, at submergence depth $z_0 = -300$ m, in an isovelocity $c = 1500$ m/s ocean acoustic waveguide of depth $h = 3000$ m. Left column result obtained by the normal mode series, keeping 40 terms: (**a**) real part, (**b**) imaginary part. Right column: field obtained from the mirror series using six terms: (**c**) real part, (**d**) imaginary part.

The same result concerning the acoustic field generated by the horizontal and vertical dipole source is presented in Figures 5 and 6, respectively.

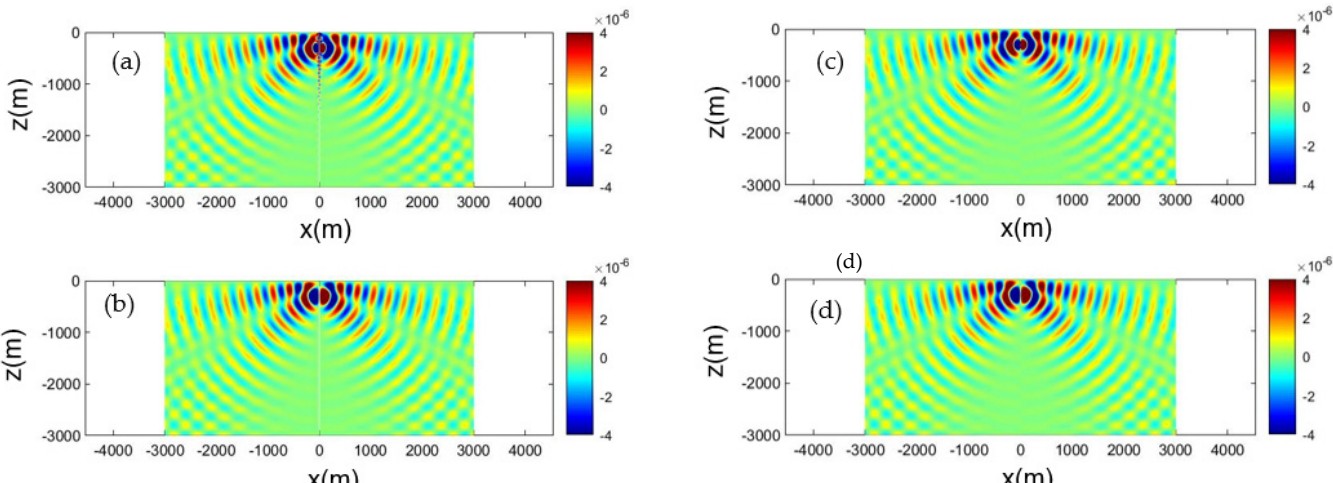

**Figure 5.** Calculated acoustic field corresponding to the horizontal dipole source, Equation (23). Source frequency 5 Hz, at submergence depth $z_0 = -300$ m, in an isovelocity $c = 1500$ m/s ocean acoustic waveguide of depth $h = 3000$ m. Left column result obtained by the normal mode series, keeping 40 terms: (**a**) real part, (**b**) imaginary part. Right column: field obtained from the mirror series using six terms: (**c**) real part, (**d**) imaginary part.

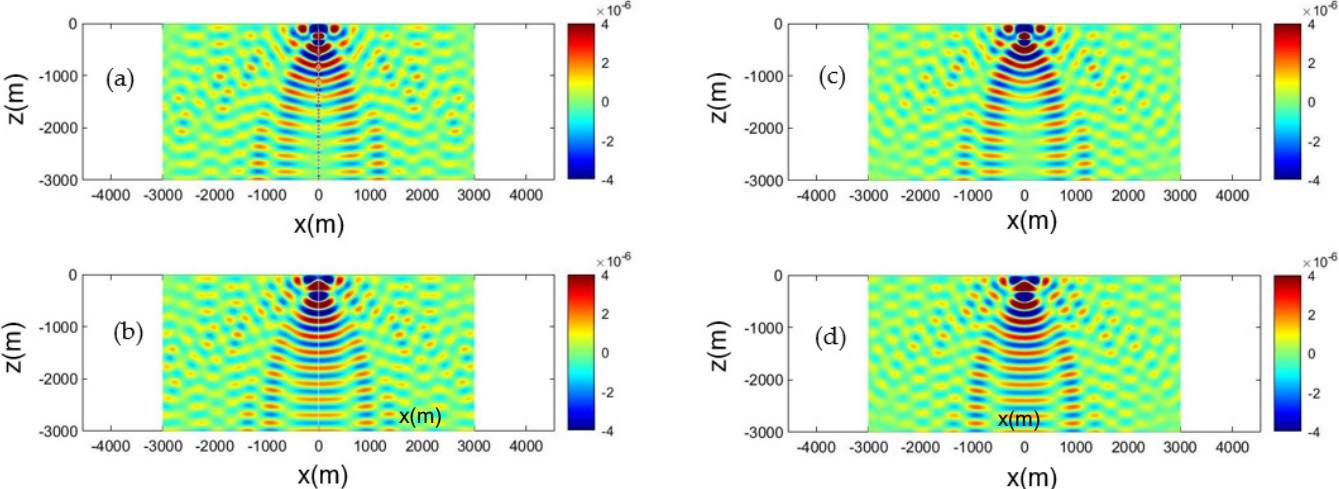

**Figure 6.** Calculated acoustic field corresponding to the vertical dipole source, Equation (24). Source frequency 5 Hz, at submergence depth $z_0 = -300$ m, in an isovelocity $c = 1500$ m/s ocean acoustic waveguide of depth $h = 3000$ m. Left column result obtained by the normal mode series, keeping 40 terms: (**a**) real part, (**b**) imaginary part. Right column: field obtained from the mirror series using six terms: (**c**) real part, (**d**) imaginary part.

## 4. Three-Dimensional Acoustic Wave Scattering Problem

### 4.1. Formulation of the Scattering Problem

Consider the acoustic field generated by a point source located at position $\mathbf{x}_0$ and propagating acoustic waves in a homogeneous isotropic medium. If the acoustic waves are scattered on the boundary $\partial D$ of an obstacle $D$, then the total acoustic field will be the sum of the incident and the scattered field. If the acoustic waves are time-harmonic with the frequency, then for the space-dependent parts in the frequency domain:

$$p(\mathbf{x}) = p_i(\mathbf{x}, \mathbf{x}_0) + p_S(\mathbf{x}) \tag{26}$$

where $p(\mathbf{x})$ is the total field, $p_i(\mathbf{x}, \mathbf{x}_0)$ is the incident field, and $p_S(\mathbf{x})$ is the scattered field. The latter satisfies the homogeneous Helmholtz equation:

$$\nabla^2 p_S(\mathbf{x}) + k^2 p_S(\mathbf{x}) = 0 \tag{27}$$

where $k$ is the wavenumber.

The mathematical description of the scattering of time-harmonic waves by an obstacle $D$ leads to boundary-value problems for the Helmholtz equation. The given data values of $p$ on the boundary $\partial D$ of the obstacle physically correspond to prescribing the acoustic pressure, while the normal derivative of $p$ on $\partial D$ physically corresponds to prescribing the normal component of the acoustic velocity. The obstacle is characterized as acoustically soft when the acoustic pressure vanishes on $\partial D$ and as acoustically hard when the normal acoustic velocity vanishes on $\partial D$. If $\mathbf{u}_i$, $\mathbf{u}_S$ are the acoustic velocities, corresponding to the incident and the scattered field, respectively, the two basic types of boundary conditions (BC) are:

$$\text{Dirichlet BC}: \ p(\mathbf{x}) = 0 \Leftrightarrow p_S(\mathbf{x}) = -p_i(\mathbf{x}, \mathbf{x}_0), \ \mathbf{x} \in \partial D \tag{28}$$

$$\text{Neumann BC}: \ \mathbf{u}(\mathbf{x}) \cdot \mathbf{n}(\mathbf{x}) = 0 \Leftrightarrow \mathbf{u}_S(\mathbf{x}) \cdot \mathbf{n}(\mathbf{x}) = -\mathbf{u}_i(\mathbf{x}, \mathbf{x}_0) \cdot \mathbf{n}(\mathbf{x}), \ \mathbf{x} \in \partial D \tag{29}$$

In Equation (29), $\mathbf{n}(\mathbf{x})$ is the unit vector normal to $\partial D$ with a direction towards the exterior of $D$. The Neumann boundary condition in Equation (29) may also be written as:

$$\mathbf{u}(\mathbf{x}) \cdot \mathbf{n}(\mathbf{x}) = 0 \Leftrightarrow \frac{\partial p_S(\mathbf{x})}{\partial n(\mathbf{x})} = -\frac{\partial p_i(\mathbf{x}, \mathbf{x}_0)}{\partial n(\mathbf{x})}, \ \mathbf{x} \in \partial D \tag{30}$$

A boundary condition that more realistically represents the acoustic properties of the obstacle is:

$$\text{Impedance BC}: \ \mathbf{u}(\mathbf{x}) \cdot \mathbf{n}(\mathbf{x}) + Z(p - p_0) = 0, \ \mathbf{x} \in \partial D \tag{31}$$

where $Z$ is the acoustic impedance of the obstacle, $p$ is the acoustic pressure and $p_0$ is the pressure of the undisturbed medium. Equation (31) states that the normal velocity on the boundary is proportional to the excess pressure on the boundary.

The problem of acoustic wave scattering consists of the calculation of the scattered field $p_S$ outside of the domain $D$. In the absence of sea surface and bottom (free field), the solution $p_S(\mathbf{x})$ should satisfy Equation (27) along with the corresponding BCs (28), (30), or (31) plus the Sommerfeld radiation condition, which completely characterizes the solutions of the Helmholtz equation at infinity. In the case of the ocean waveguide, $p_S(\mathbf{x})$ should also satisfy the following BCs at the sea surface, $z = 0$, and the sea bottom, $z = -h$:

$$\text{Dirichlet BC}: \ p_S = 0, \ z = 0 \tag{32}$$

$$\text{Neumann BC}: \ \frac{\partial p_S}{\partial n} = 0, \ z = -h \tag{33}$$

Equation (33) represents the BC if the sea bottom is considered as acoustically hard. A more accurate representation would be to take into account the acoustic properties of the sea bottom through a mixed BC, such as Equation (31).

Based on the above, the integral

$$p_S(\mathbf{x}) = \int_{\partial D} G(\mathbf{x}, \mathbf{y}) \sigma(\mathbf{y}) dS(\mathbf{y}), \ \mathbf{x} \in R^3 \backslash \partial D \tag{34}$$

with continuous density $\sigma$ is a solution of the exterior Dirichlet problem (acoustically soft obstacle) provided that $\sigma$ is a solution of the integral equation:

$$\int_{\partial D} G(\mathbf{x}, \mathbf{y}) \sigma(\mathbf{y}) dS(\mathbf{y}) = -p_i(\mathbf{x}, \mathbf{x}_0), \ \mathbf{x} \in \partial D \tag{35}$$

(see, e.g., [24]). The same solution given by (34) solves the exterior Neumann problem (acoustically hard obstacle) provided that $\sigma$ is a solution of the integral equation:

$$\frac{\sigma(\mathbf{y})}{2} - \int_{\partial D} \frac{\partial G(\mathbf{x}, \mathbf{y})}{\partial \mathbf{n}(\mathbf{x})} \sigma(\mathbf{y}) dS(\mathbf{y}) = -\frac{\partial p_i(\mathbf{x}, \mathbf{x}_0)}{\partial \mathbf{n}(\mathbf{x})}, \quad \mathbf{x} \in \partial D \tag{36}$$

In the above equations, $G(\mathbf{x}, \mathbf{y})$ is the fundamental solution of the Helmholtz equation in $R^3$ (Green's function) given by a normal series expansion, Equation (17), or by a superposition of fields of monopole sources according to the image theory.

The incident field $p_i(\mathbf{x}, \mathbf{x}_0)$ depends on the type of acoustic source. In the case of a monopole acoustic source with unit strength, $p_i(\mathbf{x}, \mathbf{x}_0)$ is given again by Equation (17). In the case of an acoustic dipole, there are two cases already mentioned in Section 3.1: one is the horizontal dipole generated by thrust, and the other is the vertical dipole generated by lift. The acoustic field in these two cases is provided by Equations (23) and (24) through a normal mode expansion. Alternatively, the effect of the sea surface and bottom on the incident field can be provided by the superposition of the elementary fields according to the image theory, as mentioned in Section 3.2. Normal mode expansion provides a better representation of the far field, whilst the method of images offers a better representation of the near field. Therefore, the second choice is preferable when the generation of noise by flapping-foils in the neighborhood of vessels is examined, which can be used as an initial solution to be propagated at long distances through a time-domain representation of the FW-H equation.

### 4.2. Acoustic Boundary Element Method

The integrals in Equations (34)–(36) are calculated numerically using the Boundary Element Method (BEM). According to BEM, the surface of the boundary $\partial D$ is discretized into a number $NTE$ of four-node quadrilaterals $E_j$:

$$\partial D = \bigcup_{j=1}^{NTE} E_j \tag{37}$$

In this way, any integral on the boundary $\partial D$ is replaced by a sum of integrals on the boundary elements $E_j$:

$$I = \int_{\partial D} \sigma(\mathbf{y}) F(\mathbf{x} - \mathbf{y}) \, dS(\mathbf{y}) = \sum_{j=1}^{NTE} \sigma_j \int_{E_j} F(\mathbf{x} - \mathbf{y}) dE_j \tag{38}$$

where $\sigma_j$ is the source distribution discretized on the elements' centers and $F(\mathbf{x} - \mathbf{y})$ can be the Green's function or its derivative.

For each element $E_j$, a transformation is defined from the Cartesian coordinate system $(x, y, z)$ to the local curvilinear system $(\xi, \eta)$ in $[-1, 1]^2$, as depicted in Figure 7. By considering a bilinear interpolation, the following relationship connects the Cartesian and the curvilinear coordinates of any integration point on $E_j$:

$$\mathbf{y} = \sum_{k=1}^{4} N_k(\xi, \eta) \, \mathbf{y}_k \tag{39}$$

where $\mathbf{y}$ is the position vector of the integration point, $\mathbf{y}_k$ is the position vector of the k-node of the element, and $N_k$ are the basis functions of the interpolation given by:

$$\begin{aligned} N_1 &= 0.25(\xi - 1)(\eta - 1), \; N_2 = -0.25(\xi + 1)(\eta - 1) \\ N_3 &= 0.25(\xi + 1)(\eta + 1), \; N_4 = -0.25(\xi - 1)(\eta - 1) \end{aligned} \tag{40}$$

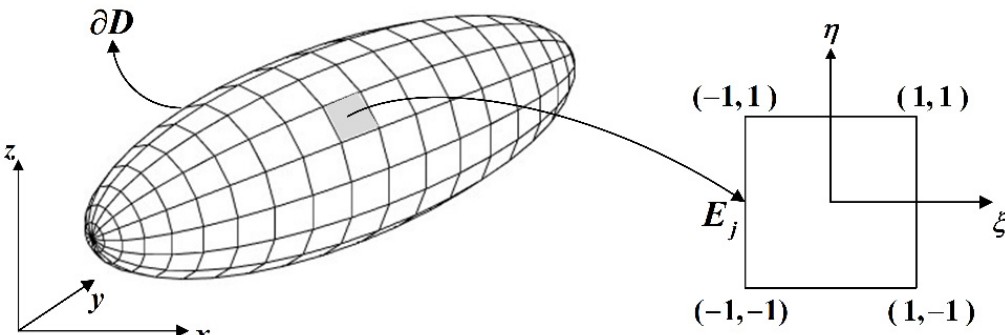

**Figure 7.** Transformation of the boundary element $E_j$ from the Cartesian coordinate system $(x, y, z)$ to the local curvilinear system $(\xi, \eta)$.

Equation (39) can also be written in the form:

$$\mathbf{y} = \mathbf{q}_1 + \mathbf{q}_2 \xi + \mathbf{q}_3 \eta + \mathbf{q}_4 \xi \eta \tag{41}$$

with:

$$\begin{aligned}
\mathbf{q}_1 &= 0.25(\ \mathbf{q}_1 + \mathbf{q}_2 + \mathbf{q}_3 + \mathbf{q}_4), & \mathbf{q}_2 &= 0.25(-\mathbf{q}_1 + \mathbf{q}_2 + \mathbf{q}_3 - \mathbf{q}_4) \\
\mathbf{q}_3 &= 0.25(-\mathbf{q}_1 - \mathbf{q}_2 + \mathbf{q}_3 + \mathbf{q}_4), & \mathbf{q}_4 &= 0.25(\ \mathbf{q}_1 - \mathbf{q}_2 + \mathbf{q}_3 + \mathbf{q}_4)
\end{aligned} \tag{42}$$

Equation (41) is used for the transformation of the kernel $F(\mathbf{x} - \mathbf{y})$ to the $(\xi, \eta)$ coordinate system and its numerical integration over $E_j$.

The elementary surface $dE_j$ is transformed into the $(\xi, \eta)$ coordinate system using the relationship:

$$dE_j = |\mathbf{e}_1 \times \mathbf{e}_2| d\xi \, d\eta, \quad \mathbf{e}_1 = \partial \mathbf{y} / \partial \xi, \quad \mathbf{e}_2 = \partial \mathbf{y} / \partial \eta \tag{43}$$

where $\mathbf{e}_1$, $\mathbf{e}_2$ are the vectors tangential to the surface and $\mathbf{e}_1 \times \mathbf{e}_2$ is the vector normal to the surface (Figure 8). Combining the definitions of $\mathbf{e}_1$, $\mathbf{e}_2$ with Equation (41), the vector $\nu = \mathbf{e}_1 \times \mathbf{e}_2$ is expressed as:

$$\nu = |\mathbf{e}_1 \times \mathbf{e}_2| = (\mathbf{q}_2 \times \mathbf{q}_3) + (\mathbf{q}_3 \times \mathbf{q}_4)\xi + (\mathbf{q}_4 \times \mathbf{q}_3)\eta \tag{44}$$

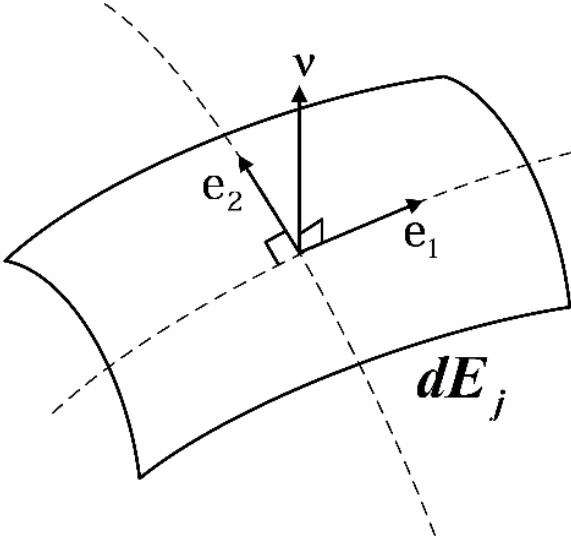

**Figure 8.** Vectors $\mathbf{e}_1$, $\mathbf{e}_2$ tangential to the elementary surface $dE_j$ and the normal vector $\nu = \mathbf{e}_1 \times \mathbf{e}_2$.

Therefore, the integral in Equation (38) is written as:

$$I = \sum_{j=1}^{NTE} \sigma_j \int_{-1}^{1} \int_{-1}^{1} F(\xi, \eta) |(\mathbf{q}_2 \times \mathbf{q}_3) + (\mathbf{q}_3 \times \mathbf{q}_4)\xi + (\mathbf{q}_4 \times \mathbf{q}_3)\eta | d\xi d\eta \qquad (45)$$

The double integral of (45) represents the contribution of the $j$ element to the control point **y**. When the integral Equations (35) and (36) are solved numerically, the centers of the elements are used as the control points. Satisfying the integral equations at all control points results in a system of $NTE$ equations with $NTE$ unknowns, the discretized sources $\sigma_i$, $i = 1, NTE$. In matrix form, the resulting system of equations for the Dirichlet problem can be written as:

$$[\sigma_1 \, \sigma_2 \ldots \sigma_{NTE}] \begin{bmatrix} A_{11} & A_{12} & \cdots & A_{1,NTE} \\ A_{21} & \cdots & \cdots & \cdots \\ \cdots & \cdots & \cdots & \cdots \\ A_{NTE,1} & \cdots & \cdots & A_{NTE,NTE} \end{bmatrix} = \begin{bmatrix} b_1 \\ b_2 \\ \cdots \\ b_{NTE} \end{bmatrix} \qquad (46)$$

where:

$$A_{ij} = \int_{-1}^{1} \int_{-1}^{1} F(\xi, \eta) |(\mathbf{q}_2 \times \mathbf{q}_3) + (\mathbf{q}_3 \times \mathbf{q}_4)\xi + (\mathbf{q}_4 \times \mathbf{q}_3)\eta | d\xi d\eta \qquad (47)$$

is the contribution coefficient of the $j$ element to the $i$ control point, $b_i = -p_i(\mathbf{x}_i, \mathbf{x}_0)$ is the incoming field, and $F$ stands for the Green's function transformed in $(\xi, \eta)$ coordinates. For the Neumann problem, the resulting system of equations will be:

$$\frac{1}{2} \begin{bmatrix} \sigma_1 \\ \sigma_2 \\ \cdots \\ \sigma_{NTE} \end{bmatrix} + [\sigma_1 \, \sigma_2 \ldots \sigma_{NTE}] \begin{bmatrix} A_{11} & A_{12} & \cdots & A_{1,NTE} \\ A_{21} & \cdots & \cdots & \cdots \\ \cdots & \cdots & \cdots & \cdots \\ A_{NTE,1} & \cdots & \cdots & A_{NTE,NTE} \end{bmatrix} = \begin{bmatrix} b_1 \\ b_2 \\ \cdots \\ b_{NTE} \end{bmatrix} \qquad (48)$$

In the contribution coefficients $A_{ij}$, $F$ is now the derivative of the Green's function, $\partial G / \partial n$, transformed in $(\xi, \eta)$ coordinates and $b_i = -\partial p_i(\mathbf{x}_i, \mathbf{x}_0) / \partial n(\mathbf{x}_i)$.

## 5. Numerical Results and Discussion

In this section, the acoustic field predicted by the solution of the 3D scattering problem using the developed BEM method is presented in the case of a sound source located close to the stern of an ellipsoid modeling a small underwater vehicle, in an isovelocity $c = 1500$ m/s ocean acoustic waveguide of depth $h = 200$ m. The submergence depth is $z_0 = -30$ m and the horizontal distance of the source from the stern is $0.1\alpha$, where $\alpha$ is the major semi-axis of the ellipsoid. The cases of a monopole, horizontal dipole, and vertical dipole source at a frequency of 100 Hz are considered. Figures 9–12 depict the acoustic field when the ellipsoid is acoustically hard, whilst Figures 13–16 depict the acoustic field when the ellipsoid is acoustically soft. The sea bottom is considered rigid in all calculations. The results include the real part and the modulus of the calculated acoustic field in two planes: (i) a vertical xz-plane passing from the source position and cutting through the major axis of the ellipsoid, and (ii) a transverse yz-plane passing through the source position in front of the vessel's stern. A number of $20 \times 30 = 600$ elements have been used for the discretization of the ellipsoid in the BEM method, which has been proven to be enough for the numerical convergence of the results at the considered source frequency.

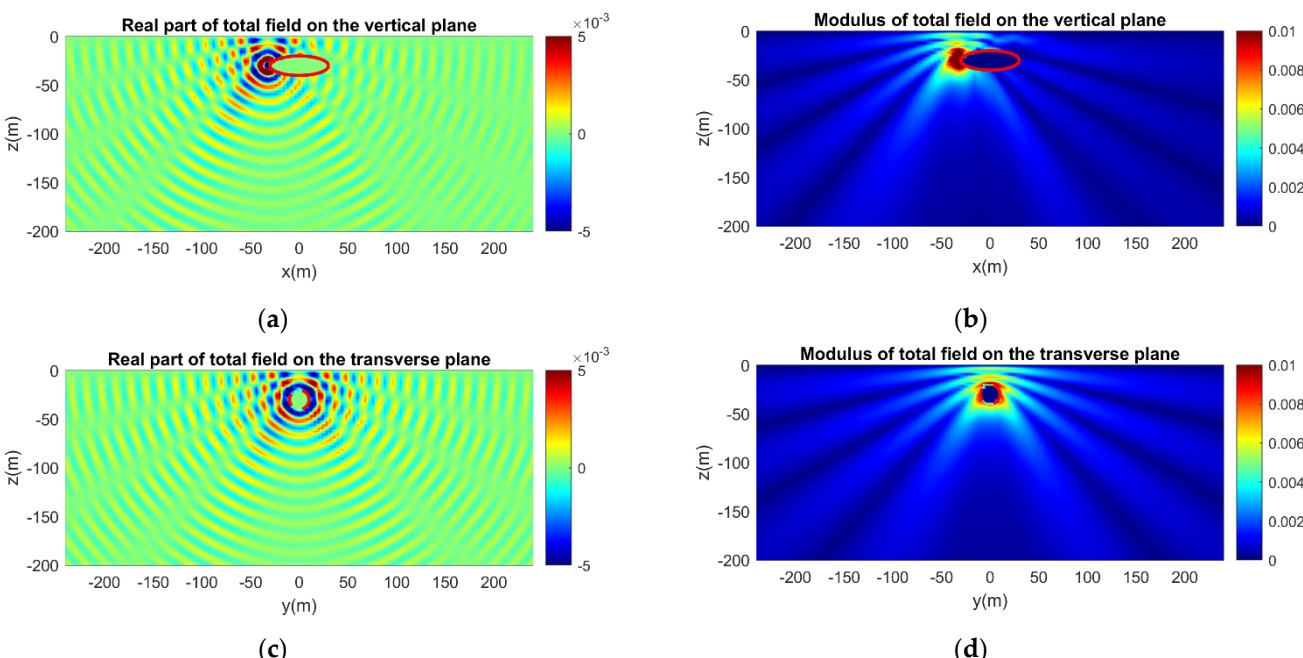

**Figure 9.** Calculated acoustic field in the semi-infinite ocean waveguide considering scattering of a monopole source field by an acoustically hard ellipsoid. The real part and the modulus of the field are presented along the vertical plane passing from the source position and cutting through the major axis of the ellipsoid and on the transverse plane passing from the source position in front of the ellipsoid. (**a**) Real part and (**b**) modulus of the acoustic field on the vertical plane. (**c**) Real part and (**d**) modulus of the acoustic field on the transverse plane.

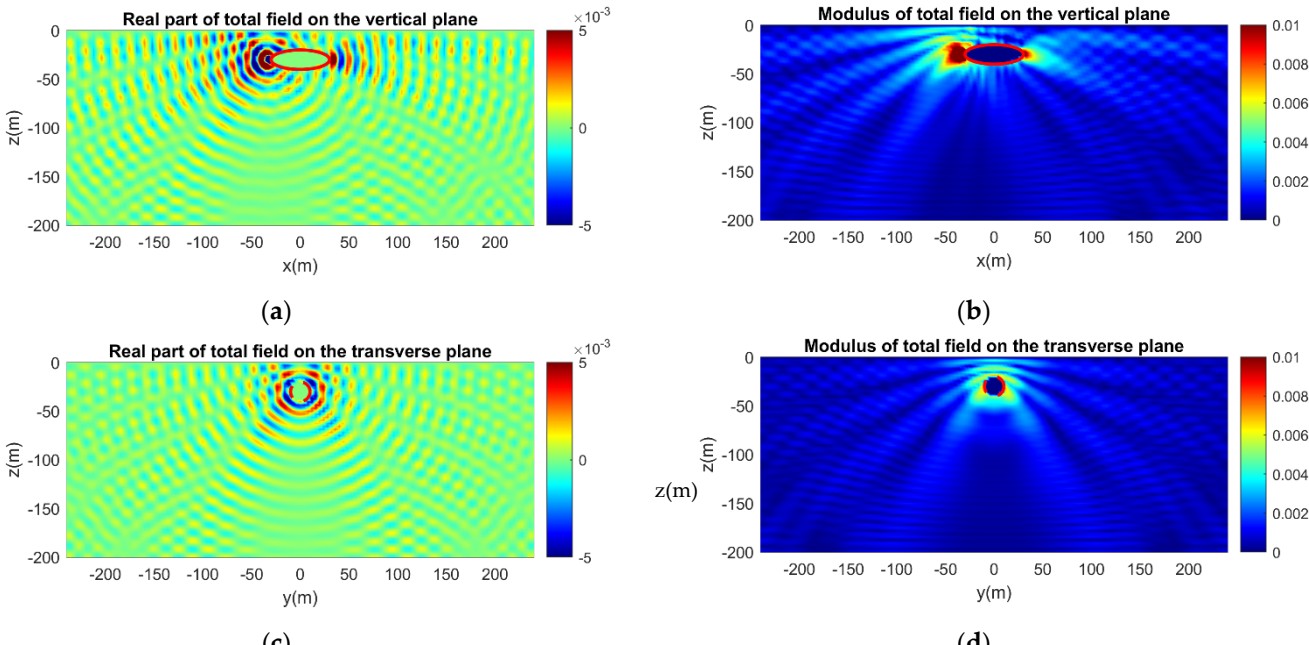

**Figure 10.** Calculated acoustic field in the ocean waveguide considering scattering of a monopole source field by an acoustically hard ellipsoid, with the effect of rigid sea bottom at depth *h* = 200 m. (**a**) Real part and (**b**) modulus of the acoustic field on the vertical plane. (**c**) Real part and (**d**) modulus of the acoustic field on the transverse plane.

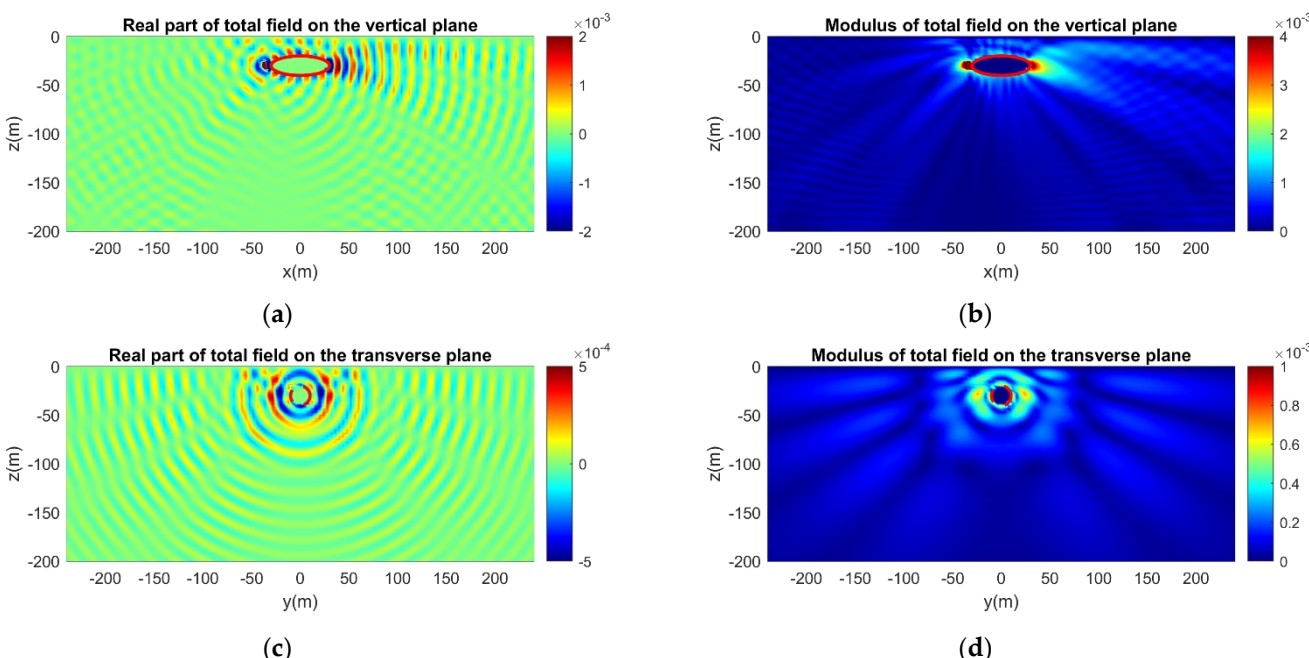

**Figure 11.** Same as Figure 10, but for a horizontal dipole source. The axis of the dipole is parallel to the major axis of the ellipsoid (x-axis). (**a**) Real part and (**b**) modulus of the acoustic field on the vertical plane. (**c**) Real part and (**d**) modulus of the acoustic field on the transverse plane.

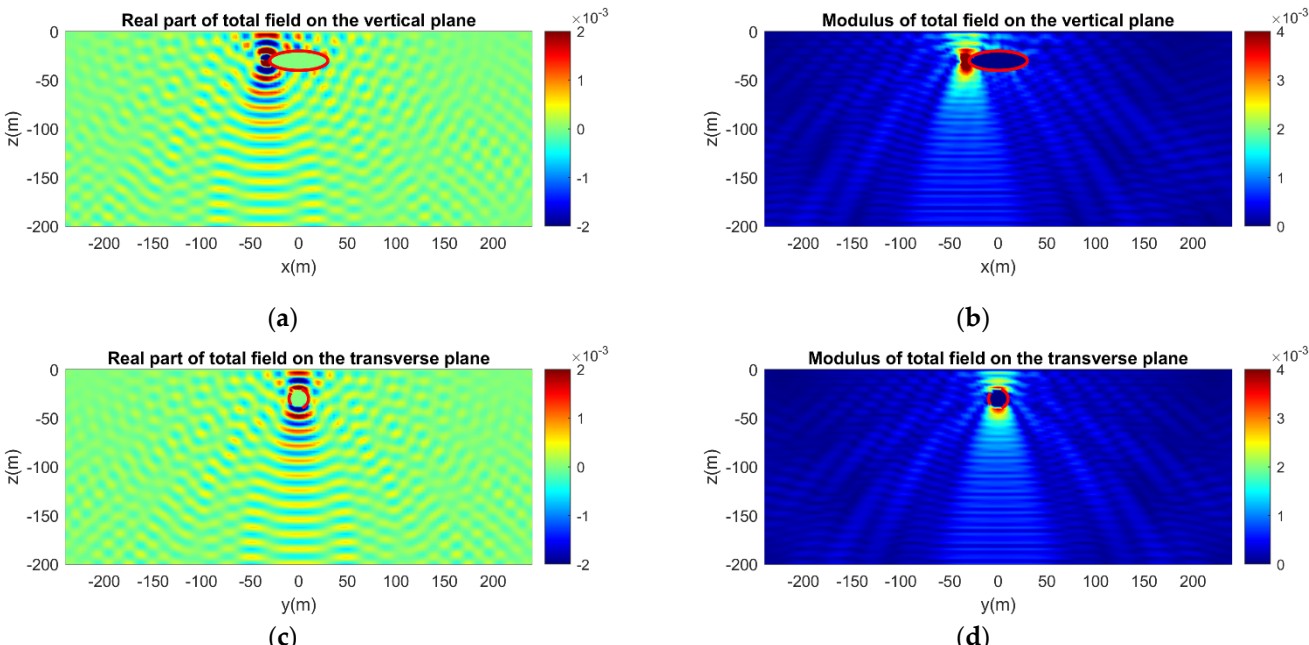

**Figure 12.** Same as Figures 10 and 11, but for a vertical dipole source. The axis of the dipole is parallel to the z-axis. (**a**) Real part and (**b**) modulus of the acoustic field on the vertical plane. (**c**) Real part and (**d**) modulus of the acoustic field on the transverse plane.

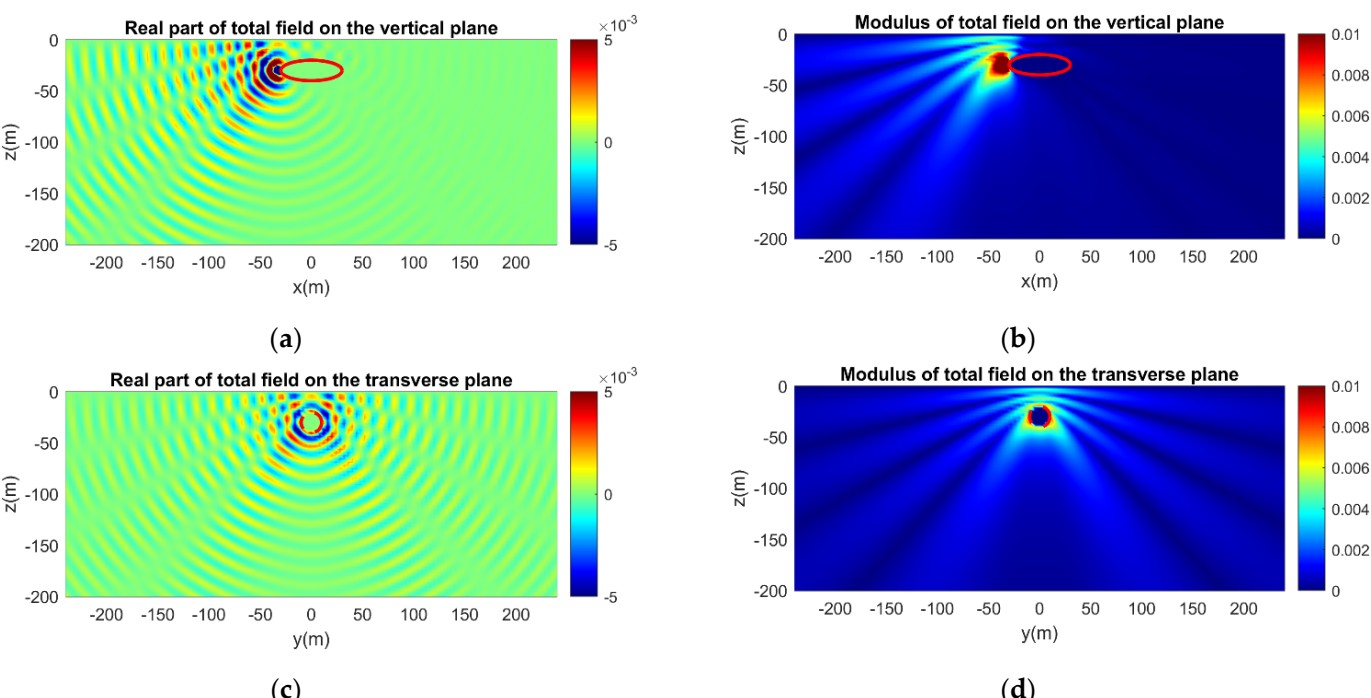

**Figure 13.** Calculated acoustic field in the semi-infinite ocean waveguide considering scattering of a monopole source field by an acoustically soft ellipsoid. (**a**) Real part and (**b**) modulus of the acoustic field on the vertical plane. (**c**) Real part and (**d**) modulus of the acoustic field on the transverse plane.

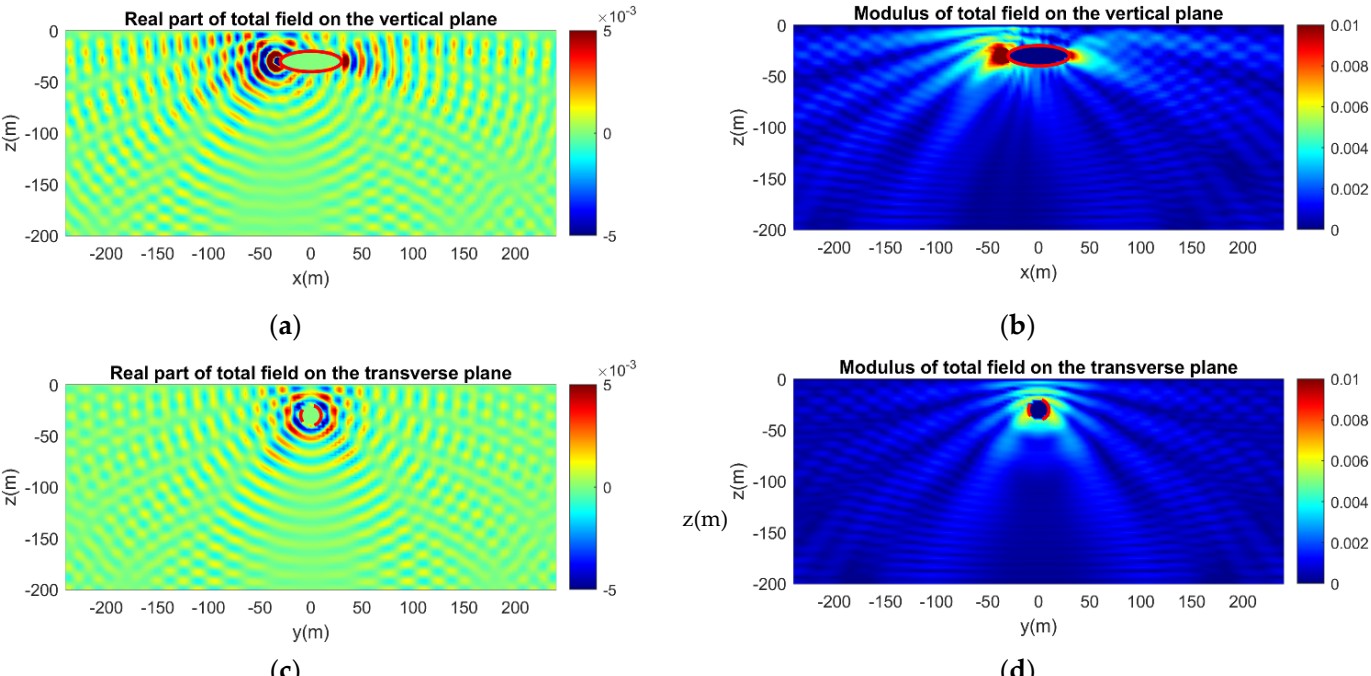

**Figure 14.** Calculated acoustic field in the ocean waveguide considering scattering of a monopole source field by an acoustically soft ellipsoid, with the effect of rigid sea bottom at depth *h* = 200 m. (**a**) Real part and (**b**) modulus of the acoustic field on the vertical plane. (**c**) Real part and (**d**) modulus of the acoustic field on the transverse plane.

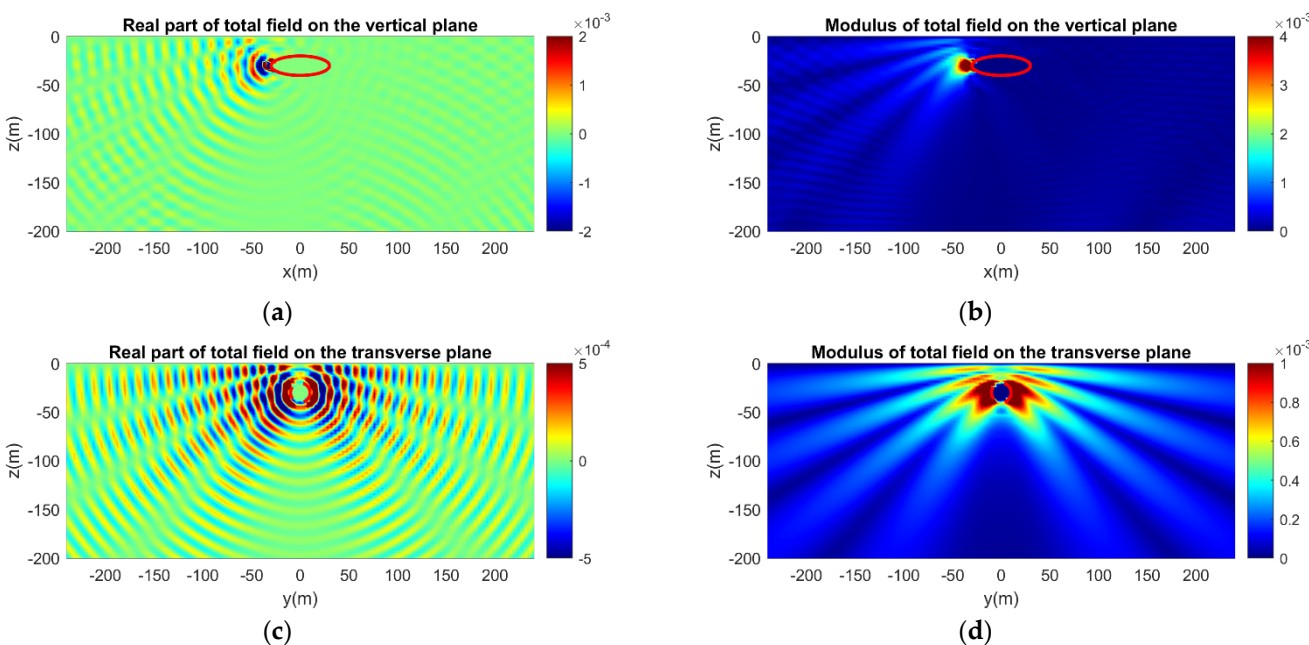

**Figure 15.** Same as Figure 14, but for a horizontal dipole source. The axis of the dipole is parallel to the major axis of the ellipsoid (x-axis). (**a**) Real part and (**b**) modulus of the acoustic field on the vertical plane. (**c**) Real part and (**d**) modulus of the acoustic field on the transverse plane.

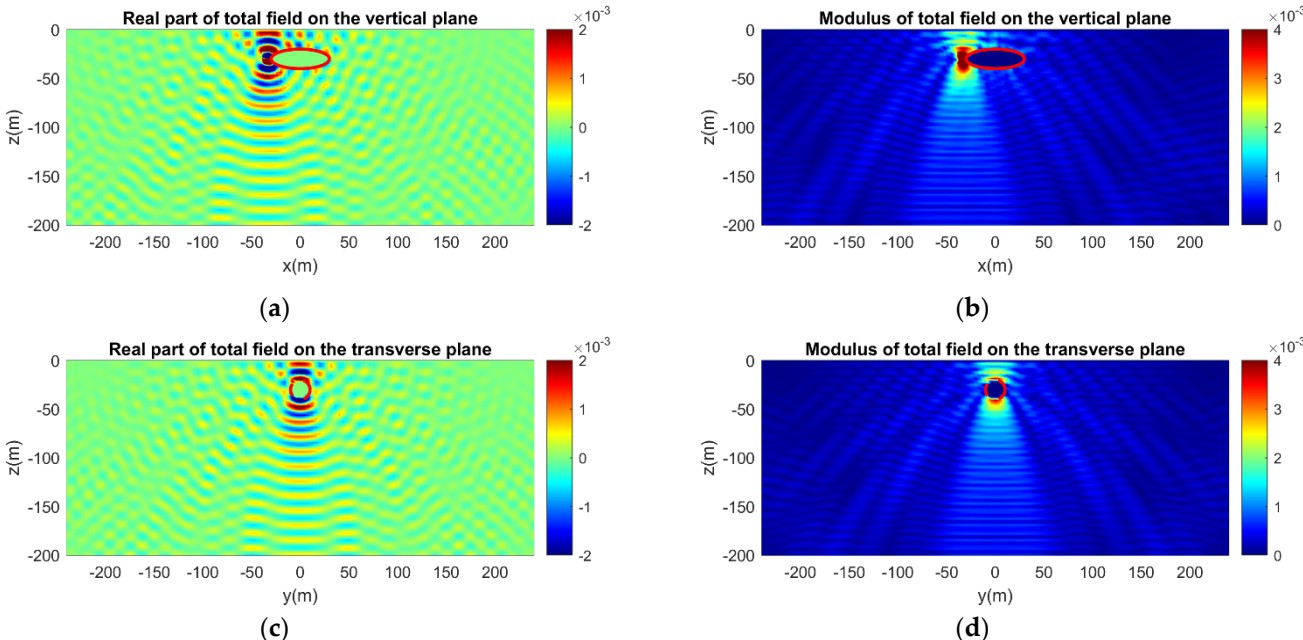

**Figure 16.** Same as in Figure 14, but for a vertical dipole source. The axis of the dipole is parallel to the z-axis. (**a**) Real part and (**b**) modulus of the acoustic field on the vertical plane. (**c**) Real part and (**d**) modulus of the acoustic field on the transverse plane.

In Figure 9, the acoustic field for a monopole source is presented initially for the scattering problem in the vertically semi-infinite waveguide. The comparison with Figure 10 reveals the effect of the rigid sea bottom as the reflected waves disturb the initial smooth pattern of the acoustic field and reinforce the scattering effect around the vessel prow. Induced variations are evident in both the real part and the modulus of the acoustic field.

In Figure 11, the excitation comes from a horizontal dipole source and the two-lobe pattern in the region close to the source is reproduced. In general, the sound levels are

lower compared to those of the monopole sound source; therefore, a different scale is implemented in order to properly depict the sound variations. In the transverse plane, the incident field from the horizontal dipole is zero and the acoustic field is only composed of the scattered field and the sea bottom reflections. Additionally, as shown in the vertical plane results, the sound reflected by the sea bottom right under the dipole position is negligible. As a result, the sound levels are significantly lower and the pattern is rather smooth in the transverse plane. In Figure 12, the excitation comes from a vertical dipole source and now the directivity of the two-lobe pattern changes, displaying maximum radiation in the vertical direction. The effect of the rigid sea bottom is, thus, increased and is reflected with more intense variations of both the real part and the modulus in the direction of the dipole axis.

Figures 13–16 correspond to the same cases as Figures 9–12, but the ellipsoid is now acoustically soft. Again, the calculated acoustic field is initially presented for the semi-infinite waveguide when the excitation is a monopole source (Figure 13). Compared to the acoustically hard ellipsoid (Figures 9–12), the effect of scattering is less intense, as depicted in both the real part and the modulus of the calculated acoustic field around the ellipsoid. This is justified by the acoustic behavior of the soft ellipsoid. Similar results are obtained for the horizontal and the vertical dipole sources (Figures 15 and 16). Again, the presence of the acoustically soft ellipsoid results in less scattering and less influence on the total sound field. In both types of dipole sources, the two-lobe pattern is distinguishable, exhibiting the directivity along the dipole axis.

The effect of the vehicle hull on the directivity of generated noise is clearly observed by comparing the field of the monopole source presented in Figures 9a and 10a, in the case of a hard and soft hull boundary, respectively, with Figure 4a, concerning the same result without the presence of the vehicle. Similarly, the changes due to the scattering effect of the vessel in the directivity characteristics of noise from the corresponding dipole sources can be observed by comparing the field presented in Figure 11a with Figure 5a for the horizontal dipole source, and in Figure 12a with Figure 6a concerning the vertical dipole, respectively.

Moreover, an important consequence of the free-surface boundary condition which is included in the present formulation is the Lloyd Mirror effect that is clearly depicted in Figures 9, 11 and 14 in the subplots illustrating the total acoustic field on the vertical plane emitted by the monopole, horizontal, and vertical dipole sources, respectively.

## 6. Conclusions

In this work, we consider the generation of noise by biomimetic flapping thrusters that are used for the propulsion of small marine vessels and AUVs and their directivity pattern, including the scattering, free-surface, and seabed effects, which are important for the characterization of the acoustic spectra to be subsequently used for sound propagation in the underwater sea environment. Focusing on the low-frequency band, the generating monopole and dipole source terms of the FW-H equation associated with the hydrodynamic noise by the biomimetic thruster are considered, providing the corresponding data for the Helmholtz equation for treating the three-dimensional scattering problem in the ocean acoustic waveguide in the frequency domain. Next, a 3D BEM model is developed and applied to calculate the scattering field, which together with the previous source field data provides us with the complete acoustic pattern in the neighborhood of the vessel. This could be further exploited for calculating the propagation of the acoustic field to far distances in the waveguide and evaluate the noise footprint of such systems in the sea environment. The numerical results are presented in selected cases, illustrating that the vessel hull geometry and acoustic properties of its surface, as well as the sea surface and seabed effects, are important for the determination of the directionality of the generated noise, which could significantly affect the long-range propagation to various azimuthal directions in the underwater ocean environment. Future work will be directed towards the inclusion of the effects of various other important parameters, such as the medium inhomogeneity, the variable bottom topography, and the seabed acoustic properties.

**Author Contributions:** This work was supervised by K.B. The BEM numerical scheme was developed by J.P. The draft of the text was prepared by all authors and the numerical simulations were handled by J.P. and I.M. Conceptualization, K.B.; methodology, K.B. and J.P.; software, J.P.; writing—original draft preparation, J.P. and I.M.; writing—review and editing, J.P. and I.M.; supervision, K.B. All authors have read and agreed to the published version of the manuscript.

**Funding:** The present work has been supported by the Seatech H2020 project. The APC have been offered by JMSE.

**Institutional Review Board Statement:** Not applicable.

**Informed Consent Statement:** Not applicable.

**Data Availability Statement:** Not applicable.

**Acknowledgments:** The present work has been supported by Seatech H2020 project received funding from the European Union's Horizon 2020 research and innovation program under the grant agreement No 857840. The opinions expressed in this document reflect only the author's view and in no way reflect the European Commission's opinions. The European Commission is not responsible for any use that may be made of the information it contains.

**Conflicts of Interest:** The authors declare no conflict of interest.

## Appendix A. Calculation of the Flapping-Foil Hydrodynamic Loads by 3D BEM

A vortex ring element method based on quadrilateral elements will be used to discretize the 3D flapping wing and its trailing vortex wake, and the singularity strengths are calculated to satisfy directly the no-entrance boundary condition on the surface of the foil, along with the Kutta condition; see Belibassakis and Malefaki [16]. In this method, the exact boundary condition is satisfied on the actual wing surface, in contrast with lifting surface models where the boundary condition is satisfied on the mean camber surface and the thickness effects are taken into account using the linearization procedure and a corresponding source-sink lattice. The solution is based on a time-stepping technique, and at the beginning of the motion, only the bound vortex ring elements on the unsteady thruster exist. The closing segment of the trailing-edge vortex elements represents the starting vortex. At the first time step, there are no wake panels. During the second time step, the wing is moved along its flight path and each trailing-edge vortex panel sheds a wake panel with a vortex strength equal to its circulation in the previous time step. This time-step methodology can be continued for any type of foil path, and at each time step, the vortex wake corner points can be moved by the local velocity, so that the wake rollup can be simulated (see also [25]).

The problem is solved by calculating the influence coefficients of the induced potential and velocity $F_{ij}$, $\mathbf{U}_{ij} = (U_{ij}, V_{ij})$, $for\ i, j = 1, \ldots K$, induced by each vortex ring element on each collocation point on the wing, which is selected as the centroid of each quadrilateral element. The latter quantities are used to set up a linear system of equations by constructing the coefficient matrix. In this respect, the flow tangency condition is implemented on the wing surface, requiring zero normal velocity. Consequently, in the present case, the discrete system of equations expressing the flow tangency condition at the collocation points on the wing is:

$$\sum_{k=1}^{K} A_{lk}\Gamma_k = \mathbf{b}_l - \mathbf{n}_l \cdot \sum_{k=1}^{K_w(t)} U_{lk}^w \Gamma_{kl}^w, \text{ for } l = 1, 2.., K \tag{A1}$$

where $\Gamma_K$ denotes the bound vortex ring element strengths and the matrix coefficient is composed by $A_{ij} = \mathbf{n}_i \mathbf{U}_{ij}$, $for\ i = 1, \ldots K$, and $\mathbf{n}_i$, $i = 1, .., K$ is the unit normal vector directed to the exterior of the body. The index $K_w(t) = M \times N_w(t)$ corresponds to the number of wake panels generated by the unsteady wing motion up to the time instant $t$, where $N_w(t) = t/\delta t$, and $\delta t$ is the time step. The system is supplemented by a Morino-type

Kutta condition used to determine the vortex ring intensity in the wake element adjacent to the trailing edge. The first term in the right-hand side of Equation (A1) is defined by:

$$\mathbf{b}_k = -\mathbf{u}_k \cdot \mathbf{n}_k, \ k = 1, ..K \tag{A2}$$

with $\mathbf{u}_k$ denoting the relative flow velocity at the collocation points of the wing:

$$\mathbf{u} = (u_x, u_y, u_z) = U\mathbf{i} - \frac{d\theta}{dt}(\mathbf{j} \times \mathbf{r}_w) - \frac{dh}{dt}\mathbf{k} \tag{A3}$$

where $\mathbf{i}, \mathbf{j}, \mathbf{k}$, are the unit vectors along the axes $x$, $y$, and $z$, respectively, and $\mathbf{r}_w$ denotes the position vector on the wing. After obtaining the velocity, the pressure distribution is calculated by applying the unsteady Bernoulli's equation providing the instantaneous distribution of the pressure coefficient:

$$C_p = \frac{p - p_\infty}{1/2\rho U^2} = 1 - \frac{|\mathbf{w}|^2}{U^2} - \frac{2}{U^2}\frac{\partial \Phi}{\partial t} \tag{A4}$$

and finally, time-dependent hydrodynamic responses concerning flapping thruster forces and moments are calculated by pressure integration over the wing surface.

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
