# Peer review of "Scattering and Directionality Effects of Noise Generation from Flapping Thrusters Used for Propulsion of Small Ocean Vehicles"

_jmse, doi:10.3390/jmse10081129_

Round 1

Reviewer 1 Report

Comments and Suggestions for Authors
r:

Noise propagation in marine environment is an important topic in academic and engineering. The authors theoretically researched the generation of noise by flapping-foils and the acoustic field characteristic based on BEM and FW-H equation. More works should be done before considering publication.

Questions and suggestions:

1the authors emphasized on the importance of flapping-foils in the background elaboration. However, the introduction about the noise theory and its research method are relatively less. It is suggested to introduce the latest research advances about noise.

2the paper cited an important paper (Reference 14), the method totally referenced the previous research results. The BEM numerical scheme and the numerical simulations are not clear. It is suggested to provide the numerical and simulation details.

3there are some spell mistakes in the paper, I would recommend the authors check the spell and grammar carefully.

Reviewer 2 Report

Please see attached document, 
